# Cross-Lingual Cross-Target Stance Detection with Dual Knowledge Distillation Framework

**Ruike Zhang[1,2], Hanxuan Yang[2,1], Wenji Mao[1,2]***

[1]Institute of Automation, Chinese Academy of Sciences
[2]School of Artificial Intelligence, University of Chinese Academy of Sciences
{zhangruike2020,yanghanxuan2020,wenji.mao}@ia.ac.cn

## Abstract

Stance detection aims to identify the user's attitude toward specific *targets* from text, which is an important research area in text mining and benefits a variety of application domains. Existing studies on stance detection were conducted mainly in English. Due to the low-resource problem in most non-English languages, cross-lingual stance detection was proposed to transfer knowledge from high-resource (source) language to low-resource (target) language. However, previous research has ignored the practical issue of no labeled training data available in target language. Moreover, target inconsistency in cross-lingual stance detection brings about the additional issue of unseen targets in target language, which in essence requires the transfer of both language and target-oriented knowledge from source to target language. To tackle these challenging issues, in this paper, we propose the new task of cross-lingual cross-target stance detection and develop the first computational work with dual knowledge distillation. Our proposed framework designs a cross-lingual teacher and a cross-target teacher using the source language data and a dual distillation process that transfers the two types of knowledge to target language. To bridge the target discrepancy between languages, cross-target teacher mines target category information and generalizes it to the unseen targets in target language via category-oriented learning. Experimental results on multilingual stance datasets demonstrate the effectiveness of our method compared to the competitive baselines [1].

## 1 Introduction

Stance detection aims to automatically identify the user's attitude (e.g., "in favor of" or "against") towards specific *targets* (e.g., entities, topics or claims) (Augenstein et al., 2016), which is an important research area in text mining and social me-

dia analytics. It has been applied to diverse domains, such as public attitude mining, market analysis (Küçük and Can, 2020), veracity prediction (Wei et al., 2019) and many others.

Existing work on stance detection has mainly focused on monolingual setting and been conducted on English datasets (Küçük and Can, 2020; Schiller et al., 2021; Li et al., 2021; Zhang et al., 2022; Huang et al., 2023). In contrast to English, most other languages lack abundant annotated data for training quality stance detection models. To address the low-resource problem and improve model performance, cross-lingual stance detection has been proposed to transfer knowledge from high-resource (source) language to low-resource (target) language. Mohtarami et al. (2019) propose a contrastive language adaptation method to align representations in two languages, which relies on the labeled training data in target language. Hardalov et al. (2022) transfer the knowledge from English to target languages through pre-training and prompt-tuning for few-shot (and zero-shot) cross-lingual stance detection (Vamvas and Sennrich, 2020) with few (and no) training data in target languages.

None of these works has addressed the practical issue when there are *only unlabeled training data* available in target language, which is a prevalent phenomenon in many low-resource languages. Consequently, the typical method based on supervised contrastive learning (Mohtarami et al., 2019) is not workable due to the lack of supervised signals for target language, and pre-training & prompt-tuning method (Hardalov et al., 2022) cannot fully utilize the unlabeled training data in target languages as well. Since labeled data are extremely scarce in low-resource languages, these unlabeled data are the valuable resource of target domain that can be utilized to facilitate various tasks in general, and in particular, we utilize them to help bridge cross-lingual gap in our work.

In addition, cross-lingual stance detection usu-

---

*Corresponding author

[1] Source code: https://github.com/ALUKErnel/CCSD.

ally brings about target inconsistency problem, that is, the occurrences and distributions of the concerned targets may vary considerably across languages due to the differences in socio-cultural backgrounds and linguistic expressions. Without the labeled data, target inconsistency causes the additional issue of *unseen targets* in target language, which in essence requires the transfer of both language-related and target-oriented knowledge from source to target language.

To address the above issues, we consider cross-lingual distillation (Xu and Yang, 2017) as a good fit to train a teacher model on the source language data so as to acquire the pseudo supervised signals for a student model trained on the unlabeled target language data. Since the language and target-oriented knowledge transferring from source to target language is of different types, dual teacher models are required for our purpose. Previously, multi-teacher distillation in language processing (Li et al., 2022) was typically used to enhance the transfer of the same type of knowledge. In contrast, our aim is to transfer different types of cross-language and cross-target knowledge, which needs to consider not only the design of individual teacher models but also their combination scheme. In particular, cross-target teacher model is built on the basis of cross-lingual teacher model, and relies on the fine-grained target knowledge, in which semantically correlated targets can be aggregated to indicate category information and utilized to generalize for the unseen targets.

In this paper, we propose the new task of **cross-lingual cross-target stance detection**, which is fundamentally different from both cross-target stance detection task in monolingual setting and existing cross-lingual stance detection task. We develop a **C**ross-lingual **C**ross-target **S**tance **D**etection (CCSD) method via designing a dual knowledge distillation framework for this task, which trains a cross-lingual teacher and a cross-target teacher on the source language data and distills the two types of knowledge to the student model. To bridge target inconsistency between languages, *cross-target teacher* first learns correlated target representations to mine the category information and fuses the aggregated semantic representations for refinement. It then devises category-oriented contrastive learning to enhance the generalization ability of the cross-target model. *Cross-lingual teacher* is a multilingual language model

which is prompt-tuned with cross-lingual templates. To reduce the impact of language differences on cross-target knowledge distillation, cross-lingual teacher model functions as the initialized encoder for cross-target teacher. Finally, in the dual distillation process, cross-lingual teacher and cross-target teacher jointly produce the pseudo-labels for the student model, and distill the two types of knowledge to the unlabeled target language data with varying degrees of target inconsistency control.

The contributions of our work are as follows:

- To tackle the issues of unlabeled data and unseen targets in target language, we identify the new task of cross-lingual cross-target stance detection, and make the first attempt to propose an integrated dual teacher-student distillation framework.

- To bridge the target inconsistency gap, cross-target teacher mines the category information via target representation learning and refinement, and generalizes it to the unseen targets via category-oriented contrastive learning.

- We conduct experiments on multilingual stance datasets with varying target settings and the results demonstrate the effectiveness of our proposed method compared to the competitive baselines.

## 2  Related Work

**Monolingual Stance Detection**  Stance detection has been well studied on English datasets (Mohammad et al., 2016; Sobhani et al., 2017; Conforti et al., 2020; Allaway and Mckeown, 2020; Schiller et al., 2021). The mainstream research is mainly as follows: (1) *Stance detection for pre-defined targets*, which trains a classifier for several pre-defined targets. Previous methods mainly learn target-specific representations with attention (Du et al., 2017; Wei et al., 2018; Siddiqua et al., 2019; Sun et al., 2018; Li and Caragea, 2019).

(2) *Cross-target stance detection*, which transfers knowledge from one target to another related target. Existing work learns target-independent representations (Augenstein et al., 2016; Xu et al., 2018) and mines transferrable features such as topic words (Wei and Mao, 2019) and semantic-emotion knowledge (Zhang et al., 2020).

(3) *Zero/few-shot stance detection*, which transfers knowledge from known targets to unseen tar-

gets with zero/few training data. Existing methods learn target-invariant representations with adversarial training (Allaway et al., 2021) or mine transferable information across targets with clustering (Allaway and Mckeown, 2020) or contrastive learning based methods (Liang et al., 2022a,b). Other methods introduce external knowledge such as Wikipedia knowledge and commonsense knowledge from ConceptNet to enhance zero/few-shot stance detection (Zhu et al., 2022; Liu et al., 2021).

There is also other work that conducts multi-dataset learning (Hardalov et al., 2021; Li et al., 2021) for cross-domain stance detection.

**Cross-Lingual Stance Detection** Compared to the abundant data resources in English, there are scarce datasets in other low-resource languages (Xu et al., 2016; Lozhnikov et al., 2018; Khouja, 2020; Cignarella et al., 2020; Martínez et al., 2023). To tackle the low-resource problem in non-English languages, cross-lingual stance detection is proposed to transfer knowledge from source language to target language (Mohtarami et al., 2019; Küçük and Can, 2020). Some studies address this problem via constructing multilingual stance datasets and baseline methods (Taulé et al., 2017; Lai et al., 2020; Zotova et al., 2020; Vamvas and Sennrich, 2020; Agerri et al., 2021; Barriere et al., 2022). Others develop identification methods for cross-lingual stance detection (Mohtarami et al., 2019; Hardalov et al., 2022), which we detail below.

Mohtarami et al. (2019) first propose a contrastive language adaptation method to align representations in two languages, by encouraging samples with the same label in different languages to be close in the embedding space. However, their method is not workable when there are no labeled data in target language. Vamvas and Sennrich (2020) further propose zero-shot cross-lingual stance detection in the setting of no training data in target language. To tackle this problem setting, Hardalov et al. (2022) pre-train XLM-R with additional sentiment-based corpora and transfer the knowledge to target languages with prompt-tuning for few-shot (and zero-shot) cross-lingual stance detection. However, their work overlooks the availability of unlabeled data in target languages, which can be utilized to further bridge the language gap in cross-lingual stance detection.

Therefore, our work concentrates on the setting of cross-lingual stance detection when there are no labeled training data available in target language. It is not only advantageous to alleviate the difficulty of data annotation in low-resource languages, but also beneficial to the utilization of unlabeled training data in target language. Moreover, none of the above methods considers the target inconsistency problem caused by cross-lingual stance detection, where additional cross-target knowledge transfer is required. In this paper, we identify the important research theme of cross-target stance detection in cross-lingual setting, which can be viewed as an integrated task of cross-lingual and cross-target stance detection, and an integrated computational framework is required to address knowledge transfer at both language and target levels with combination scheme.

## 3 Proposed Method

Figure 1 shows the overall structure of the proposed cross-lingual cross-target stance detection method CCSD. We first train a cross-lingual teacher and a cross-target teacher on the source language data, and then distill the learned knowledge to the student model trained with unlabeled data in target language. *Cross-lingual teacher* is a multilingual pre-trained language model which is prompt-tuned with cross-lingual templates and consistency constraints to enhance cross-lingual ability solely with the source language data. *Cross-target teacher* learns target representations to mine the target category with highly-correlated targets, and then generalizes the target category to the unseen targets in target language through contrastive learning.

### 3.1 Task Definition

We denote the training data in source language as $D_{src} = \{(t_i, c_i), y_i\}_{i=1}^{N_s}$, where $t_i, c_i$ are the $i$-th target and text, $y_i$ is the stance label, and $N_s$ is the number of samples in $D_{src}$. The target set of $D_{src}$ is represented as $T_{src} = \{\bar{t}_i\}_{i=1}^{n_s}$, where $\bar{t}_i$ is the $i$-th unique target in $D_{src}$ and $n_s$ is the number of targets. Similarly, we denote the **unlabeled** training data in target language as $D_{tgt} = \{(t'_i, c'_i)\}_{i=1}^{N_t}$ with the target set $T_{tgt} = \{\bar{t}'_i\}_{i=1}^{n_t}$. Generally, $N_t \ll N_s$ in cross-lingual stance detection. We train two teacher models on $D_{src}$ and distill knowledge to the student model trained on $D_{tgt}$, then predict stance labels on the test set in target language.

### 3.2 Cross-Lingual Teacher

We adopt mBERT as cross-lingual teacher and design cross-lingual stance templates for prompt-

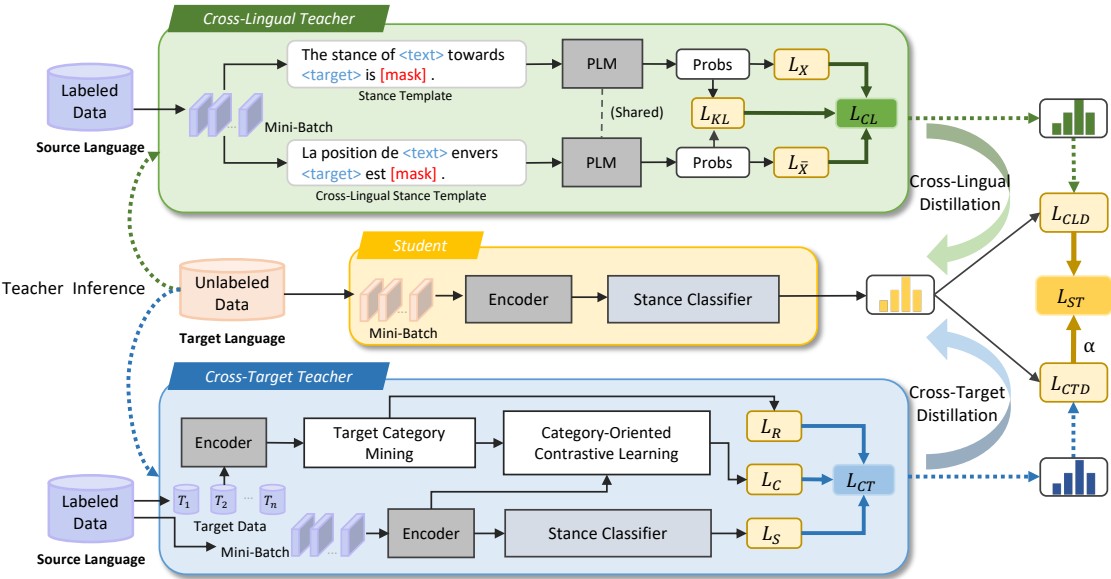

Figure 1: The overall architecture of our proposed cross-lingual cross-target stance detection method CCSD with dual knowledge distillation.

tuning inspired by Qi et al. (2022) to enhance the cross-lingual ability of the model solely with the source language data. The *template* designed for stance detection is presented in English as: "*The stance of <text> towards <target> is [MASK].*", where "<target>" and "<text>" are the slots to fill the input target and text in, "[MASK]" is the token for predicting the stance label and others are *prompts* for assisting the language model. For each source language, we translate the prompt words to get the monolingual template $X$, and then translate the prompt words into its target language to get cross-lingual template $\overline{X}$. The verbalizer maps stance labels into specific words, and we set "favor" and "against" for the favor and against stance labels respectively. We feed the target-text pair with both monolingual and cross-lingual templates into mBERT and get the hidden vector of "[MASK]" $\boldsymbol{h}^{[\text{MASK}]} \in \mathbb{R}^d$ in $X$ and $\overline{\boldsymbol{h}}^{[\text{MASK}]}$ in $\overline{X}$. The predicted distribution is calculated as follows:

$$\hat{\boldsymbol{y}}_i^X = \text{Softmax}(\boldsymbol{W}_{lm}\boldsymbol{h}_i^{[\text{MASK}]}) \quad (1)$$

$$\hat{\boldsymbol{y}}_i^{\overline{X}} = \text{Softmax}(\boldsymbol{W}_{lm}\overline{\boldsymbol{h}}_i^{[\text{MASK}]}) \quad (2)$$

where $\boldsymbol{W}_{lm} \in \mathbb{R}^{l \times d}$ is the projection matrix which transforms the predicted hidden vector into specific words, and $l$ is the size of the vocabulary. We minimize the cross-entropy loss $\mathcal{L}_X = \text{CrossEntropy}(\boldsymbol{y}_i^X, \hat{\boldsymbol{y}}_i^X)$ on $X$ and $\mathcal{L}_{\overline{X}} = \text{CrossEntropy}(\boldsymbol{y}_i^{\overline{X}}, \hat{\boldsymbol{y}}_i^{\overline{X}})$ on $\overline{X}$, where $\boldsymbol{y}_i^X$ ($\boldsymbol{y}_i^{\overline{X}}$) is the one-hot vector transformed from the stance label with the verbalizer.

To further enhance the cross-lingual ability, we add constraints to the predicted distributions based on monolingual and cross-lingual templates following Qi et al. (2022). Specifically, we force the predicted distributions based on $X$ and $\overline{X}$ as close as possible. We use Kullback-Leibler divergence to measure the two distributions and minimize the consistency loss function $\mathcal{L}_{KL}$:

$$\mathcal{L}_{KL} = \frac{1}{N_s} \sum_{i=1}^{N_s} (\text{KL}(\hat{\boldsymbol{y}}_i^X \| \hat{\boldsymbol{y}}_i^{\overline{X}}) + \text{KL}(\hat{\boldsymbol{y}}_i^{\overline{X}} \| \hat{\boldsymbol{y}}_i^X))$$
$$(3)$$

Finally, we optimize cross-lingual teacher with the combined loss $\mathcal{L}_{CL}$:

$$\mathcal{L}_{CL} = \mathcal{L}_X + \mathcal{L}_{\overline{X}} + \mathcal{L}_{KL} \quad (4)$$

### 3.3 Cross-Target Teacher

Figure 2 illustrates the structure of the proposed cross-target teacher. To bridge the target inconsistency between the source and target languages, cross-target teacher mines target category through clustering on the target representations. Specifically, it learns target representations with stance-related association modeling and fuses the semantic information to refine target representations. Then, category-oriented contrastive learning is devised to generalize the category information to the unseen targets in target language.

#### 3.3.1 Encoder Module

We use mBERT as the encoder module and initialize it with the prompt-tuned cross-lingual teacher

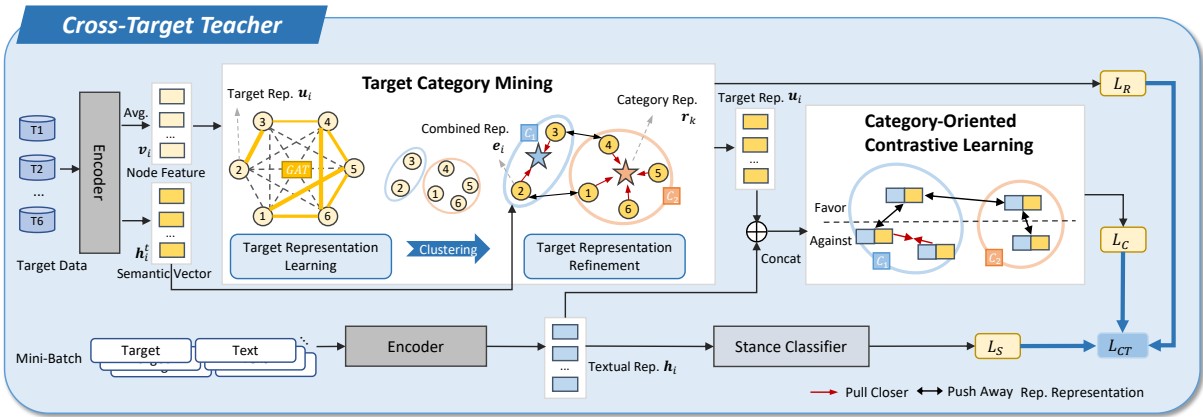

Figure 2: The architecture of our proposed cross-target teacher.

to reduce the impact of language differences on the cross-target knowledge transfer. Specifically, we feed the target $t_i$ and text $c_i$ into the encoder with special tokens and take the output of "[CLS]" as the textual representation $\boldsymbol{h}_i \in \mathbb{R}^d$:

$$\boldsymbol{h}_i = \text{mBERT}([CLS]t_i[SEP]c_i[SEP]) \quad (5)$$

where $t_i$ and $c_i$ are the target and text in source language respectively.

### 3.3.2 Target Category Mining

We mine target category information in the aspects of the stance-related associations and semantics similarity. Specifically, we learn target representations by modeling target associations with graph, and cluster the target representations into target categories. We further refine the target representations by fusing the target semantic information.

**Target Representation Learning** We construct target graph $\mathcal{G} = \langle \mathcal{V}, \mathcal{A} \rangle$ to learn the structural target representations, where $\mathcal{V} = (\boldsymbol{v}_1, \dots, \boldsymbol{v}_{n_s})^\top \in \mathbb{R}^{n_s \times d}$ represents the node features and $\mathcal{A} = (\mathcal{A}_{i,j}) \in \{0, 1\}^{n_s \times n_s}$ is the adjacency matrix. Each target in source language is treated as a node in the graph. Intuitively, the stance-related association between targets is reflected by their corresponding sample set. Thus, for the $i$-th target $\bar{t}_i$ in source language, we also use mBERT to encode each sample with $\bar{t}_i$ to get $\widetilde{\boldsymbol{h}}_{i,k} = \text{mBERT}([CLS]\bar{t}_i[SEP]c_k[SEP])$, where $c_k$ is the text of the $k$-th sample with target $\bar{t}_i$, and calculate the average vector as the node feature $\boldsymbol{v}_i = \frac{1}{N_i} \sum_{k=1}^{N_i} \widetilde{\boldsymbol{h}}_{i,k}$, where $N_i$ the number of samples with target $\bar{t}_i$. Assuming that there are associations between all pairs of targets in the beginning, we build a fully connected graph and set the adjacency matrix $\mathcal{A}_{i,j} = \mathcal{A}_{j,i} = 1$.

To dynamically model the associations between targets and learn structural target representations, we feed the node features $\mathcal{V}$ and the adjacency matrix $\mathcal{A}$ into Graph Attention Network (GAT) (Veličković et al., 2018), and derive target representations $\mathcal{U} = \{\boldsymbol{u}_i\}_{i=1}^{n_s}$, where $\boldsymbol{u}_i$ is the target representation for the $i$-th target $\bar{t}_i$.

$$\mathcal{U} = \text{GAT}(\mathcal{V}, \mathcal{A}) \quad (6)$$

**Target Representation Refinement** To mine the target category with highly-associated targets, we use k-means to cluster the learned target representations $\mathcal{U}$ into $K$ categories $C = \{C_k\}_{k=1}^{K}$, where $C_k$ is the index set of targets contained in the $k$-th cluster. We then fuse the target semantic information into categories to refine the target representations and category division. Specifically, we use mBERT to derive the target semantic vector $\boldsymbol{h}_i^t \in \mathbb{R}^d$:

$$\boldsymbol{h}_i^t = \text{mBERT}([CLS]\bar{t}_i[SEP]) \quad (7)$$

Then $\boldsymbol{h}_i^t$ is concatenated with the target representation $\boldsymbol{u}_i$ to get the combined representation $\boldsymbol{e}_i = [\boldsymbol{u}_i; \boldsymbol{h}_i^t] \in \mathbb{R}^{2d}$. For each category $C_k$, we average the combined representations within the category to calculate the category representation $\boldsymbol{r}_k \in \mathbb{R}^{2d}$:

$$\boldsymbol{r}_k = \frac{1}{n_k} \sum_{i \in C_k} \boldsymbol{e}_i \quad (8)$$

where $n_k$ is the number of targets within the category $C_k$. We increase the closeness within the categories and the discrimination between categories, so as to strengthen the category division and optimize the target representation. To this end, we devise the following intra-category constraint to pull the combined representation closer to its

corresponding category, and the inter-category constraint to push away the representations from different categories:

$$\mathcal{L}_R = \sum_{i=1}^{n_s} \left( \|e_i - r_{ca_i}\|^2 - \frac{1}{n_i'} \sum_{j \notin C_i} \|e_i - e_j\|^2 \right) \quad (9)$$

where $ca_i$ is the category of $e_i$, $r_{ca_i}$ is the category representation of $e_i$, and $n_i'$ is the number of targets in other categories except $ca_i$.

### 3.3.3 Category-Oriented Contrastive Learning

To generalize the learned category information to the unseen targets in target language, we devise a category-oriented contrastive learning method. First, the textual representation $h_i$ is concatenated with its corresponding structural target representation $u_k$ to get the target-enhanced representations $z_i = [h_i; u_k] \in \mathbb{R}^{2d}$. For each anchor $z_i$ in the mini-batch, we select the samples with the same category and stance label as the positive samples and treat others as negative samples. We pull the positive samples closer and push the negative samples away by minimizing the following contrastive loss $\mathcal{L}_C$:

$$\mathcal{L}_C = -\frac{1}{N_b} \sum_{i=1}^{N_b} \frac{1}{N_b'} \sum_{j=1}^{N_b'} \psi(i,j) \mathbb{I}(y_i = y_j) \cdot$$
$$\log \frac{\mathbb{I}(i \neq j) \exp(f(z_i, z_j)/\tau)}{\sum_{k=1}^{N_b} \mathbb{I}(i \neq k) \exp(f(z_i, z_k)/\tau)} \quad (10)$$

$$\psi(i,j) = \begin{cases} 1 & \text{if } ca_i = ca_j \text{ or } t_i = t_j \\ 0 & \text{otherwise} \end{cases} \quad (11)$$

where $t_i$ and $ca_i$ are the target and category of $z_i$ respectively, $\tau$ is the temperature hyperparameter, and $N_b$, $N_b'$ are the numbers of samples and the selected positive samples in the mini-batch.

The predicted stance label $\hat{y}_i$ is obtained by feeding the textual representation $h_i$ into a stance classifier, which is a two-layer feed-forward network with a Softmax function. We adopt cross-entropy loss $\mathcal{L}_S = \text{CrossEntropy}(y_i, \hat{y}_i)$ to optimize the classifier. Finally, cross-target teacher is trained with the source language data by minimizing the combined loss $\mathcal{L}_{CT}$:

$$\mathcal{L}_{CT} = \mathcal{L}_R + \mathcal{L}_C + \mathcal{L}_S \quad (12)$$

### 3.4 Cross-Lingual Cross-Target Distillation

To transfer the cross-lingual and cross-target knowledge to target language, we devise a dual knowledge distillation process, as shown in Figure 1. The student model is trained on the unlabeled target language data with the supervisory signals produced by cross-lingual teacher and cross-target teacher respectively. Specifically, the student model is composed of an encoder module and a stance classifier, and the forward calculation is as follows:

$$h_i' = \text{mBERT}([CLS]t_i'[SEP]c_i'[SEP]) \quad (13)$$

$$\hat{y}_i' = \text{Softmax}(\text{FFN}_S(h_i')) \quad (14)$$

where $\hat{y}_i$ is the stance label predicted by the student model, and $\text{FFN}_S(\cdot)$ is the two-layer feed-forward network in the student model.

Given the target-text pair in target language, we input it together with the prompts into cross-lingual teacher and generate the supervisory signal $\hat{y}_i^{CL}$. Cross-target teacher also produces the supervisory signal $\hat{y}_i^{CT}$. We train the student model with the following cross-lingual distillation loss $\mathcal{L}_{CLD}$ and cross-target distillation loss $\mathcal{L}_{CTD}$:

$$\mathcal{L}_{CLD} = -\frac{1}{N_t} \sum_{i=1}^{N_t} (\hat{y}_i^{CL})^\top \log \hat{y}_i' \quad (15)$$

$$\mathcal{L}_{CTD} = -\frac{1}{N_t} \sum_{i=1}^{N_t} (\hat{y}_i^{CT})^\top \log \hat{y}_i' \quad (16)$$

To tackle various degrees of target inconsistency, we set a trade-off hyperparameter $\alpha$ to adjust the proportions of the cross-lingual knowledge and cross-target knowledge. Finally, the student model is optimized by minimizing the combined loss $\mathcal{L}_{ST}$:

$$\mathcal{L}_{ST} = \mathcal{L}_{CLD} + \alpha \mathcal{L}_{CTD} \quad (17)$$

## 4 Experiments

### 4.1 Datasets and Experimental Settings

**Dataset and Target Settings** X-stance (Vamvas and Sennrich, 2020) is a multilingual stance dataset on Swiss politics, where German is used as the source language and French is the target language. Each sample consists of a voter's question and a candidate's answer, and the stance can be classified into "*favor*" or "*against*". We construct two datasets from X-stance: (1) **Politics (P)** is comprised of all the data in domains "*Foreign Policy*" and "*Immigration*", with 31 different targets in total. There are 7064 samples in German and 2582 samples in French; (2) **Society (S)** consists of all the data in domains "*Society*" and "*Security*", with 32 different targets in total. It contains 7362 samples in German and 2467 samples in French.

Further, we establish three different target settings on the two datasets: (1) **All**: All the targets in source language and target language are the same; (2) **Partial**: Part of the targets in source language and target language are the same. We randomly select 50% overlap of all the targets between source and target languages; (3) **None**: None of the targets in source language and target language are the same. We randomly select 50% of the targets in source language and remaining 50% for target language. Thus, we get 6 different experimental settings (i.e., 2 datasets with 3 target settings) to verify the effectiveness of our method.

**SemEval2016** (Mohammad et al., 2016) is an English stance dataset on Twitter, which contains 4163 samples with 5 targets "*Atheism*", "*Climate Change is a real Concern*", "*Feminist Movement*", "*Hillary Clinton*" and "*Legalization of Abortion*". We use its "*favor*" and "*against*" samples as the target language data, and German data in Politics (P) and Society (S) as the source language data respectively, to establish two additional experimental settings "P-Sem-None" and "S-Sem-None".

**R-ita** is an Italian dataset of multilingual political corpus (Lai et al., 2020), which contains 833 samples about "*Constitutional Reform*". All the data are randomly split into the training and test set with a proportion of 80%-20%. Similar to SemEval2016, "*favor*" and "*against*" samples are used as the target language data and German data in Politics (P) and Society (S) are the source language data respectively, which forms two additional experimental settings in Italian "P-Rita-None" and "S-Rita-None".

**Czech** (Hercig et al., 2017) contains 1455 samples in Czech with targets of "*Miloš Zeman*" and "*Smoking Ban in Restaurants*". We split all the data into the training set and the test set with a proportion of 80%-20% randomly. In the same way, it also contributes two additional experimental settings "P-Czech-None" and "S-Czech-None".

**Implementation Details** We provide more details on the datasets and experiments in Appendices A and B.

### 4.2 Comparative Baselines

We choose the following monolingual methods as the comparative baselines. For a fair comparison, we adapt these methods to cross-lingual stance detection by replacing the original word embeddings with the hidden vectors of mBERT. (1) **TAN**

(Du et al., 2017) learns target-specific representations with an attention mechanism; (2) **BiCond** (Augenstein et al., 2016) incorporates target information into text representations with bidirectional conditional LSTMs; (3) **CrossNet** (Xu et al., 2018) uses self-attention to learn target-independent representations for cross-target stance detection; (4) **JointCL** (Liang et al., 2022b) devises target-aware prototypical graph contrastive learning for zero-shot stance detection.

We also select the methods for cross-lingual tasks to compare with our proposed method CCSD. (1) **ADAN** (Chen et al., 2018) aligns representations in the source and target languages with adversarial training; (2) **CLKD** (Xu and Yang, 2017) trains the source classifier with the labeled source language data and distills knowledge to the target model; (3) **mBERT-FT** (Devlin et al., 2019) fine-tunes the language model mBERT on the training data; (4) **mBERT-PT** prompt-tunes mBERT with stance template in source language.

### 4.3 Main Results

We use Accuracy and macro $F_1$ of "*favor*" and "*against*" as the evaluation metrics. Tables 1 and 2 give the experimental results of our method CCSD and the comparison baselines on the four datasets. It can be seen that our method basically outperforms the baseline methods in all settings, which benefits from the dual distillation of cross-lingual and cross-target knowledge and the adjustment of the proportion of these two types of knowledge in different target inconsistency cases. Especially in "Politics-None", "S-Sem-None" and "S-Rita-None", our method outperforms the comparative methods by 7.99%, 2.99% and 2.96% on $F_1$. This verifies the effectiveness of our proposed method on cross-lingual cross-target stance detection.

We can see that cross-lingual methods perform better than monolingual methods in general, indicating the importance of knowledge transfer across languages for cross-lingual stance detection. For monolingual methods, JointCL devises prototypical graph to bridge the gap between the known targets and unknown targets, and performs better than attention-based methods TAN and CrossNet, showing the effectiveness of mining high-level features to bridge the target inconsistency gap. For cross-lingual methods, fine-tuning and prompt-tuning mBERT achieve rather good results, which benefits from the cross-lingual ability of multilingual pre-

| Method | Politics | | | | | | Politics + Others | | | | | |
| | Politics-All | | Politics-Partial | | Politics-None | | P-Sem-None | | P-Rita-None | | P-Czech-None | |
| | Acc | F$_1$ | Acc | F$_1$ | Acc | F$_1$ | Acc | F$_1$ | Acc | F$_1$ | Acc | F$_1$ |
|---|---|---|---|---|---|---|---|---|---|---|---|---|
| TAN | 61.93 | 61.27 | 59.88 | 59.44 | 55.03 | 54.51 | 55.94 | 49.49 | 60.69 | 50.34 | 53.01 | 48.89 |
| BiCond | 62.64 | 62.00 | 59.30 | 59.27 | 53.14 | 52.73 | **62.51** | 50.93 | 56.49 | 50.55 | 52.78 | 51.64 |
| CrossNet | 61.93 | 60.41 | 59.30 | 58.99 | 56.60 | 56.15 | 51.52 | 48.89 | 55.73 | 50.38 | 52.31 | 51.59 |
| JointCL | 63.65 | 62.71 | 61.05 | 60.64 | 53.77 | 53.33 | 53.78 | 50.25 | 65.84 | 52.31 | 53.94 | 53.39 |
| ADAN | 60.63 | 59.27 | 58.33 | 58.18 | 56.29 | 55.08 | 54.56 | 50.19 | 61.32 | 49.93 | 52.78 | 52.70 |
| CLKD | 67.67 | 66.54 | 66.09 | 65.80 | 53.77 | 53.17 | 59.27 | 49.59 | 60.31 | 48.36 | 52.78 | 52.45 |
| mBERT-FT | 66.38 | 66.14 | 65.89 | 65.76 | 56.29 | 55.87 | 59.76 | 50.59 | 67.94 | 51.01 | 52.78 | 52.74 |
| mBERT-PT | 67.82 | 67.76 | 66.28 | 65.90 | 57.86 | 57.48 | 61.19 | 49.99 | **69.47** | 51.57 | 54.17 | 53.73 |
| **CCSD** | **70.11**† | **69.92**† | **67.44**† | **67.32**† | **65.72**† | **65.47**† | 59.27 | **51.50** | 66.92 | **54.87**† | **56.25**† | **55.89**† |

Table 1: Experimental results of baselines and CCSD on Politics and other three datasets under three target settings. For each method, we report the average score of 5 runs in percentages. The best performances are marked in bold, and † means that our proposed method CCSD is statistically significantly better than the baselines ($p < 0.05$).

| Method | Society | | | | | | Society + Others | | | | | |
| | Society-All | | Society-Partial | | Society-None | | S-Sem-None | | S-Rita-None | | S-Czech-None | |
| | Acc | F$_1$ | Acc | F$_1$ | Acc | F$_1$ | Acc | F$_1$ | Acc | F$_1$ | Acc | F$_1$ |
|---|---|---|---|---|---|---|---|---|---|---|---|---|
| TAN | 59.61 | 58.75 | 55.75 | 55.42 | 56.79 | 54.10 | 57.21 | 50.93 | 61.83 | 47.18 | 50.69 | 49.71 |
| BiCond | 61.11 | 60.33 | 59.00 | 58.38 | 53.40 | 53.14 | 62.15 | 51.53 | 63.61 | 48.62 | 50.69 | 50.69 |
| CrossNet | 60.96 | 60.42 | 57.85 | 57.43 | 56.48 | 54.38 | 61.56 | 51.51 | 63.87 | 49.29 | 51.39 | 51.09 |
| JointCL | 61.71 | 61.61 | 61.69 | 61.11 | 58.64 | 56.58 | **63.79** | 52.66 | 62.60 | 48.80 | 52.08 | 50.50 |
| ADAN | 61.71 | 61.59 | 59.77 | 59.30 | 54.63 | 54.35 | 59.90 | 51.55 | 61.83 | 48.78 | 52.08 | 50.89 |
| CLKD | 63.06 | 62.39 | 59.96 | 58.86 | 56.79 | 56.09 | 57.44 | 52.02 | 64.12 | 50.76 | 51.39 | 49.48 |
| mBERT-FT | 66.82 | 66.28 | 65.33 | 64.40 | 60.80 | 59.66 | 56.23 | 52.97 | 61.07 | 50.08 | 51.39 | 50.36 |
| mBERT-PT | 66.82 | 66.31 | 64.37 | 63.84 | 61.73 | 59.00 | **63.79** | 50.77 | 61.07 | 51.58 | 51.39 | 51.05 |
| **CCSD**(Ours) | **67.87**† | **67.32**† | **66.09**† | **65.43**† | **62.96**† | **62.48**† | 61.04 | **55.96**† | **64.38**† | **54.54**† | **52.78**† | **52.55**† |

Table 2: Experimental results of baselines and CCSD on Society and other three datasets under three target settings.

trained language model. Besides, CLKD also has competitive performances, demonstrating the effectiveness of cross-lingual knowledge distillation with no labeled data in target language. We can also see from the left half of Tables 1 and 2 that the performance of each comparison method decreases greatly from "All" to "Partial" and "None", showing that target inconsistency between languages degrades the model performance for cross-lingual stance detection.

To further verify the effectiveness of CCSD when no targets are the same between languages, we conduct experiments using three additional datasets as the target language data. The experimental results are given in the right half of the two tables. We can see that even though both the language and target gaps are large, our method still outperforms the baseline methods in general, further demonstrating the effectiveness of our dual distillation method for cross-lingual cross-target stance detection.

### 4.4 Ablation Study

The first part of Table 3 gives the results on variants of the overall framework. It can be seen that removing cross-lingual distillation (i.e., $\mathcal{L}_{CLD}$) leads to performance dropping under three target settings, showing the importance of dual knowledge distillation for cross-lingual cross-target stance detection. In "Politics-All", we can see that removing cross-lingual teacher results in the largest performance drop in the first part, which indicates that cross-lingual knowledge plays a decisive role in the target-consistent case. In setting "Politics-None", the performance decreases more when removing cross-target teacher, demonstrating that the cross-target knowledge transfer is important for target inconsistency in cross-lingual stance detection.

The results on variants of "cross-lingual teacher" are provided in the second part. Removing the consistency constraint also leads to a performance drop, indicating the effectiveness of constricting predicted distributions based on $X$ and $\overline{X}$ to be

| Variants | Politics-All | | | | Politics-Partial | | | | Politics-None | | | |
|---|---|---|---|---|---|---|---|---|---|---|---|---|
| | Acc | $F_1$ | $\Delta$Acc | $\Delta F_1$ | Acc | $F_1$ | $\Delta$Acc | $\Delta F_1$ | Acc | $F_1$ | $\Delta$Acc | $\Delta F_1$ |
| **CCSD (Ours)** | 70.11 | 69.92 | - | - | 67.44 | 67.32 | - | - | 65.72 | 65.47 | - | - |
| **w/o** cross-lingual distillation | 69.11 | 68.84 | -1.01 | -1.08 | 66.67 | 66.47 | -0.78 | -0.85 | 64.15 | 63.69 | -1.57 | -1.78 |
| **w/o** cross-lingual teacher | 66.67 | 66.61 | -3.45 | -3.31 | 65.89 | 65.37 | -1.55 | -1.95 | 64.15 | 63.83 | -1.57 | -1.64 |
| **w/o** cross-target teacher | 69.25 | 69.09 | -0.86 | -0.83 | 67.05 | 66.57 | -0.39 | -0.76 | 63.21 | 62.24 | -2.52 | -3.24 |
| **w/o** consistency constraint | 68.75 | 68.37 | -1.36 | -1.55 | 66.09 | 66.06 | -1.35 | -1.26 | 62.26 | 62.14 | -3.46 | -3.33 |
| **w/o** cross-lingual template | 67.59 | 67.25 | -2.52 | -2.67 | 65.12 | 65.07 | -2.32 | -2.25 | 62.26 | 61.34 | -3.46 | -4.13 |
| **w/o** target refinement | 69.83 | 69.41 | -0.29 | -0.51 | 67.25 | 66.82 | -0.19 | -0.50 | 64.15 | 63.57 | -1.57 | -1.90 |
| **w/o** target category | 69.54 | 68.96 | -0.57 | -0.96 | 66.47 | 65.69 | -0.97 | -1.63 | 63.21 | 62.57 | -2.52 | -2.90 |

Table 3: Ablation results of all the variants of our proposed CCSD on Politics under three different target settings.

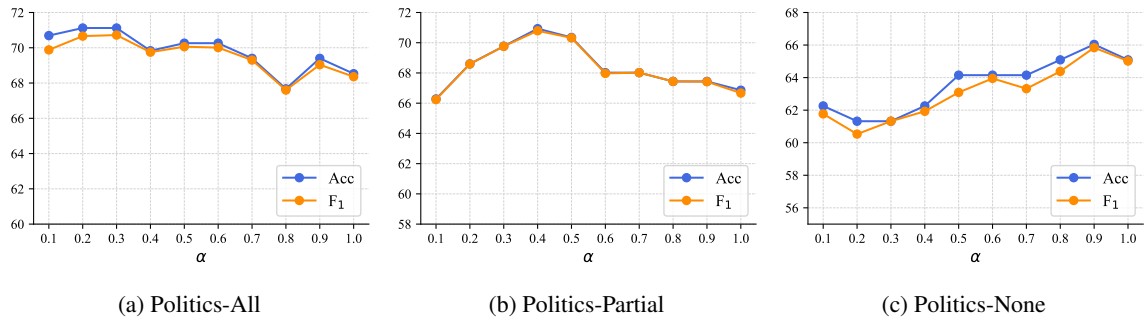

    (a) Politics-All                         (b) Politics-Partial                         (c) Politics-None

Figure 3: Impact of hyperparameters $\alpha$ in dual knowledge distillation on Politics under three different target settings.

consistent. The performance decreases more without translating the prompts into target language in "Politics-None", demonstrating that translating the prompt words is more important when none of targets are the same between languages.

The last part provides the results on variants of "cross-target teacher". Excluding target representation refinement (i.e., $\mathcal{L}_R$) causes the model to degrade more in "Politics-None", showing the refinement with semantics benefits target inconsistency. When removing the whole target category mining, the decrease is larger in "Politics-Partial" and "Politics-None", demonstrating the target categories bridge the target gap between languages and improve the generalization ability of the model.

### 4.5 Analysis on Trade-off Hyperparameter

We conduct experiments on the trade-off hyperparameter $\alpha$ which determines the proportion of the cross-lingual and cross-target knowledge in the dual distillation process. Figure 3 illustrates the impact of $\alpha$ on Politics under three different target settings. Our proposed method CCSD achieves higher performance when $\alpha$ is around 0.1~0.3, 0.4~0.5, 0.8~1.0 in "Politics-All", "Politics-Partial" and "Politics-None", respectively. Targets in the source and target languages are the same in "Politics-All",

thus it mostly relies on the target-invariant knowledge for stance detection. As the degree of target inconsistency increases, it is vital to pay more attention to the cross-target knowledge transfer in dual distillation.

## 5 Conclusion

To address target inconsistency in low-resource cross-lingual stance detection, we propose the new task of cross-lingual cross-target stance detection and develop a dual knowledge distillation framework CCSD to tackle the challenging issues of unlabeled data and unseen targets in target language. In our dual framework, a cross-lingual teacher and a cross-target teacher are trained on the source language data and distill respective knowledge to the student model trained with the unlabeled target language data. To bridge the target gap between source and target languages, cross-target teacher further mines the semantically correlated target categories, and generalizes this information to the unseen targets in target language. Experimental results on four cross-lingual datasets under varying target settings demonstrate the effectiveness of our method for cross-lingual cross-target stance detection.

## Limitations

It can be seen from the experimental results that the performance of our proposed method is less superior than some baselines in "P-Sem-None", "S-Sem-None" and "P-Rita-None" in accuracy. We speculate that this is caused by different target expressions. The targets in Politics and Society datasets are sentences of questions as shown in Tables 9 and 10. And the targets in SemEval2016 and R-ita datasets are keywords about some concepts and entities like "Atheism", "Hillary Clinton" and "Constitutional Reform". We speculate that the prompts designed for the source language cannot precisely assist in stance detection in target language with a totally different target expression. Our future work shall further explore the impact of different prompts on cross-lingual cross-target stance detection to compensate for the target discrepancy in expressions.

## Acknowledgments

This work is supported in part by the Ministry of Science and Technology of China under Grant #2022YFB2703302, and the National Natural Science Foundation of China under Grants #11832001, #71974187 and #62206287.

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

## A    More Details on Datasets

There are ten topics in X-stance including "*Economy*","*Finances*", "*Education*" and so on. We select topics "*Immigration*" and "*Foreign Policy*" to construct the "**Politics**" dataset, and "*Society*" and "*Security*" to construct the "**Society**" dataset, where both datasets have similar target-text ratios to that of the original X-stance dataset. After removing

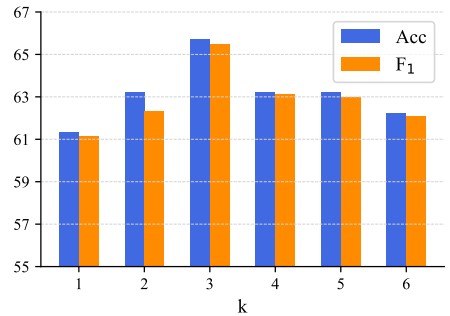

(a) Politics-None

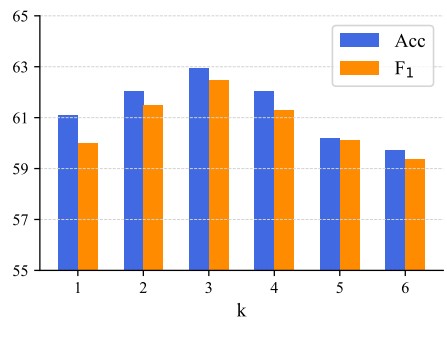

(b) Society-None

Figure 4: Impact of hyperparameters $K$ of k-means in "Politics-None" and "Society-None".

one more target in the German data than the French data, the Politics dataset contains 31 targets and the Society dataset contains 32 targets. Tables 9 and 10 show the complete list of the targets in the two datasets and their descriptions translated into English.

To verify the effectiveness for different target inconsistency cases, target settings "**All**", "**Partial**" and "**None**" are constructed, the procedures as follows: (1) **All**: All the targets in source and target languages are the same ; (2) **Partial**: We randomly select 50% of the total targets (16 for both datasets) as the overlap targets, and randomly select 8 targets from the left targets as the source-only targets. The remaining targets are served as the target-only targets; (3) **None**: We randomly select 50% of the total targets (16 for both datasets) for source language. The remaining targets are consequently served as the targets in target language.

The target sets in the source and target languages of the two datasets are shown in Tables 4 and 5. The train-test split follows that of the original X-stance, and the statistics are given in Tables 6 and 7 respectively.

| | Source Language (de) | | Target Language (fr) | | Overlap | |
|---|---|---|---|---|---|---|
| **Politics-All** | [1, 2, 3, 4, 5, 6, 7, 8, 9, 10, 11, 12, 13, 14, 15, 16, 17, 18, 19, 20, 21, 22, 23, 24, 25, 26, 27, 28, 29, 30, 31] | #31 | [1, 2, 3, 4, 5, 6, 7, 8, 9, 10, 11, 12, 13, 14, 15, 16, 17, 18, 19, 20, 21, 22, 23, 24, 25, 26, 27, 28, 29, 30, 31] | #31 | [1, 2, 3, 4, 5, 6, 7, 8, 9, 10, 11, 12, 13, 14, 15, 16, 17, 18, 19, 20, 21, 22, 23, 24, 25, 26, 27, 28, 29, 30, 31] | #31 |
| **Politics-Partial** | [1, 2, 3, 4, 5, 7, 8, 9, 11, 13, 14, 15, 16, 17, 18, 19, 20, 21, 23, 24, 25, 26, 28, 30] | #24 | [1, 4, 5, 6, 8, 10, 11, 12, 13, 14, 15, 16, 17, 18, 19, 21, 22, 23, 25, 26, 27, 29, 31] | #22 | [1, 4, 5, 8, 11, 13, 14, 15, 16, 17, 18, 19, 21, 23, 25, 26] | #16 |
| **Politics-None** | [1, 2, 6, 8, 10, 11, 12, 14, 17, 18, 20, 22, 27, 29, 30, 31] | #16 | [3, 4, 5, 7, 9, 13, 15, 16, 19, 21, 23, 24, 25, 26, 28] | #15 | [] | #0 |

Table 4: The target sets in the source and target language data in Politics.

| | Source Language (de) | | Target Language (fr) | | Overlap | |
|---|---|---|---|---|---|---|
| **Society-All** | [1, 2, 3, 4, 5, 6, 7, 8, 9, 10, 11, 12, 13, 14, 15, 16, 17, 18, 19, 20, 21, 22, 23, 24, 25, 26, 27, 28, 29, 30, 31, 32] | #32 | [1, 2, 3, 4, 5, 6, 7, 8, 9, 10, 11, 12, 13, 14, 15, 16, 17, 18, 19, 20, 21, 22, 23, 24, 25, 26, 27, 28, 29, 30, 31, 32] | #32 | [1, 2, 3, 4, 5, 6, 7, 8, 9, 10, 11, 12, 13, 14, 15, 16, 17, 18, 19, 20, 21, 22, 23, 24, 25, 26, 27, 28, 29, 30, 31, 32] | #32 |
| **Society-Partial** | [1, 3, 4, 5, 6, 7, 9, 11, 12, 13, 14, 15, 19, 20, 21, 22, 23, 25, 26, 27, 29, 30, 31, 32] | #24 | [2, 6, 7, 8, 9, 10, 11, 12, 14, 15, 16, 17, 18, 19, 20, 21, 22, 23, 24, 26, 27, 28, 29, 30] | #24 | [6, 7, 9, 11, 12, 14, 15, 19, 20, 21, 22, 23, 26, 27, 29, 30] | #16 |
| **Society-None** | [2, 4, 6, 7, 10, 11, 13, 14, 15, 17, 22, 23, 24, 25, 28, 29] | #16 | [1, 3, 5, 8, 9, 12, 16, 18, 19, 20, 21, 26, 27, 30, 31, 32] | #16 | [] | #0 |

Table 5: The target sets in the source and target language data in Society.

| Split | Politics-All | | Politics-Partial | | Politics-None | |
|---|---|---|---|---|---|---|
| | de | fr | de | fr | de | fr |
| **Train** | 5926 | 2149 | 4286 | 1543 | 3258 | 1008 |
| **Valid** | 505 | 201 | 371 | 150 | 265 | 89 |
| **Test** | 633 | 232 | 472 | 172 | 318 | 106 |

Table 6: The statistics of Politics.

| Split | Society-All | | Society-Partial | | Society-None | |
|---|---|---|---|---|---|---|
| | de | fr | de | fr | de | fr |
| **Train** | 6313 | 2051 | 5104 | 1535 | 2872 | 1135 |
| **Valid** | 487 | 194 | 357 | 145 | 241 | 90 |
| **Test** | 562 | 222 | 424 | 174 | 271 | 108 |

Table 7: The statistics of Society.

## B  Implementation Details

The proposed dual distillation framework CCSD is implemented with PyTorch and all the experiments are conducted on NVIDIA GeForce RTX 3090. We use "bert-base-multilingual-cased", which is a 12-layer, 768-hidden and 12-head model with about 110M parameters and implemented with the Transformers framework. Thus the size $d$ of textual representation derived from mBERT is 768. $K$ in k-means is set to 3. $\tau$ in category-oriented contrastive learning is 0.3. The trade-off hyperparameter $\alpha$ is 0.2 for "Politics-All", 0.5 for "Politics-Partial", 0.9 for "Politics-None"; 0.1 for "Society-All", 0.4 for "Society-Partial", 0.9 for "Society-None"; 1.0 for "P-Sem-None" and "S-Sem-None". The stance template is "*Die Haltung von <text> gegenüber*

*<target> ist [MASK].*" with German prompts and is "*La position de <text> envers <target> est [MASK].*" with French prompts. All parameters of teacher and student models are optimized by Adam (Kingma and Ba, 2015) with a learning rate of $2e^{-5}$. The batch size is 16 for cross-lingual teacher, and 32 for cross-target teacher and student model. We train all the models for 15 epochs. The running time of the whole framework is about 1 GPU hour including the training of teachers and distillation.

## C  Variants for Ablation Study

Here we provide a detailed introduction to the variants of the ablation study. Below are the variants of *overall dual distillation framework*:

- "**w/o** cross-lingual teacher distillation" means removing the cross-lingual distillation and only transferring cross-target knowledge to target language. However, cross-target teacher is still constructed based on the cross-lingual teacher.

- "**w/o** cross-target teacher" represents excluding cross-target teacher from the distillation framework.

- "**w/o** cross-lingual teacher" represents excluding cross-lingual teacher from the distillation framework.

Below are the variants of *cross-lingual teacher*:

| | Target | Text | Stance | mBERT-FT | **CCSD** |
|---|---|---|---|---|---|
| 1 | **<Consistent Target 25>** Les personnes sans-papiers devraient-elles pouvoir obtenir plus facilement un statut de séjour régularisé? | Le système doit être équitable et être mis en œuvre au cas par cas. | Against | Favor | Against |
| | **English Translation**: Should sans-papiers be able to obtain a regularized residence status more easily? | **English Translation**: The system must be fair and implemented on a case-by-case basis. | | ✗ | ✔ |
| 2 | **<Inconsistent Target 29>** La Suisse devrait-elle conclure un accord de libre-échange avec les Etats-Unis? | Trop de risques pour les standards environnementaux et sociaux... | Against | Favor | Against |
| | **English Translation**: Should Switzerland strive for a free trade agreement with the USA? | **English Translation**: Too many risks for environmental and social standards... | | ✗ | ✔ |
| 3 | **<Inconsistent Target 22>** La Confédération devrait-elle soutenir davantage les étrangères et étrangers dans leur intégration? | L'intégration est une responsabilité individuelle, mais proposer des plateformes comme des cours de langues est un pas vers une meilleure intégration. | Favor | Against | Favor |
| | **English Translation**: Should the federal government provide more support for the integration of foreigners? | **English Translation**: Integration is an individual responsibility, but offering platforms like language courses is a step towards better integration. | | ✗ | ✔ |

Table 8: Case study of predicted results with mBERT-FT and our CCSD on Politics under Partial setting

- "**w/o** consistency constraint" means removing $\mathcal{L}_{KL}$ from the combined loss $\mathcal{L}_{CL}$ for optimizing cross-lingual teacher.

- "**w/o** cross-lingual template" means removing cross-lingual template (i.e., not translating prompt words) and optimizing cross-lingual teacher solely with $\mathcal{L}_X$.

Below are the variants of *cross-target teacher*:

- "**w/o** target representation refinement" means removing $\mathcal{L}_R$ from $\mathcal{L}_{CT}$ for optimizing cross-target teacher.

- "**w/o** target category mining" means excluding the whole target category mining and removing category-related information from contrastive learning, resulting in $\mathcal{L}_C$ as follows:

$$\mathcal{L}_C = -\frac{1}{N_b}\sum_{i=1}^{N_b}\frac{1}{N_b'}\sum_{j=1}^{N_b'}\mathbb{I}(y_i = y_j)\cdot$$
$$\log\frac{\mathbb{I}(i \neq j)\exp(f(\boldsymbol{z}_i, \boldsymbol{z}_j)/\tau)}{\sum_{k=1}^{N_b}\mathbb{I}(i \neq k)\exp(f(\boldsymbol{z}_i, \boldsymbol{z}_k)/\tau)} \quad (18)$$

## D Analysis on K-Means Hyperparameter

We conduct experiments on the hyperparameter $K$ of k-means in target category mining. Figure 4 shows the impact of $K$ in "Politics-None" and "Society-None" with no targets overlapping between languages. We can see that CCSD achieves the best performance when $K$ is 3. When the number of clusters is too small, each cluster may contain more targets with low correlations, causing the model to refine target presentations mistakenly. As $K$ becomes larger, the performance gradually decreases. When the number of clusters increases, there are fewer targets in each category and fewer positive samples in contrastive learning, resulting in a decrease in the contrastive ability and generalization ability on unlabeled data.

## E Case Study

We conduct a case study to compare the predicted results of mBERT-FT and our CCSD on Politics dataset under Partial setting. Table 8 gives the comparison results of the three representative cases from the test set in the target language (i.e., French). For the consistent target in case 1, CCSD predicts the stance label correctly. Furthermore, we can see that mBERT predicts wrong stance labels on the unseen targets that are not included in the source language data (i.e., cases 2 and 3). In contrast, our CCSD enhances its generalization ability on unseen targets by target category mining and category-oriented contrastive learning so as to bridge the target inconsistency gap between source and target languages.

| ID | Domain | Target (shown below in the English form of a "topic", as given in the original dataset) |
|---|---|---|
| 1 | Immigration | Are you in favour of legalizing the status of sans papiers immigrants (i.e. immigrants who have no official paperwork) through a one-off, collective granting of residency permits? |
| 2 | Immigration | Would you support foreigners who have lived for at least ten years in Switzerland being given voting and electoral rights at municipal level throughout Switzerland? |
| 3 | Immigration | Should the state provide more funding for the integration of foreigners? |
| 4 | Immigration | Should access to "facilitated naturalization" via the Federation be made more difficult? |
| 5 | Immigration | The United Nations High Commissioner for Refugees (UNHCR) is seeking host countries for groups of refugees known as "quota refugees". Should Switzerland accept more of these groups? |
| 6 | Immigration | A popular initiative has been launched that wants to regulate immigration and thus limit migration-related population growth to 0.2% annually. Do you support this idea? |
| 7 | Foreign Policy | Would you support the introduction of the automatic exchange of bank client data between Switzerland and foreign tax authorities? |
| 8 | Foreign Policy | Should Switzerland embark on negotiations in the next four years to join the EU? |
| 9 | Foreign Policy | Should Switzerland conclude an agricultural free trade agreement with the EU? |
| 10 | Immigration | Do you support the existing agreement with the EU on the free movement of peoples? |
| 11 | Foreign Policy | Today, the Swiss Army can take part in UN or OSCE peace-keeping missions abroad, armed for self-defence purposes. Do you approve? |
| 12 | Foreign Policy | For a number of years, Switzerland has pursued a more active and open foreign policy that is less geared to strict neutrality. Do you welcome this change? |
| 13 | Foreign Policy | Should compliance with human rights play a greater role when deciding whether to enter into economic agreements with other countries (e.g. free trade agreements)? |
| 14 | Immigration | Would you support that foreigners who have lived for at least ten years in Switzerland being given voting and electoral rights at municipal level throughout Switzerland? |
| 15 | Immigration | Are you in favour of legalizing the status of sans papiers immigrants (i.e. immigrants who have no official paperwork) through a one-off, collective granting of residency permits? |
| 16 | Immigration | Do you think Switzerland should accept an increased number of refugees directly from crisis regions for which the United Nations High Commissioner for Refugees (UNHCR) needs host countries (what is called quota refugees)? |
| 17 | Foreign Policy | Should Switzerland embark on negotiations in the next four years to join the EU? |
| 18 | Foreign Policy | Should Switzerland start negotiations with the USA on a free trade agreement? |
| 19 | Foreign Policy | Should liability regulations for companies operating from Switzerland be tightened with regard to the compliance with human rights and environmental standards? |
| 20 | Foreign Policy | Do you think that Swiss foreign policy should increasingly be oriented to a strict interpretation of neutrality? |
| 21 | Foreign Policy | Should Switzerland terminate the Schengen Agreement with the EU and reintroduce increased identity checks directly on the border? |
| 22 | Immigration | Should the federal government provide more support for the integration of foreigners? |
| 23 | Immigration | Should foreigners who have lived in Switzerland for at least ten years be given the right to vote and be elected at the municipal level? |
| 24 | Immigration | Is limiting immigration more important to you than maintaining the bilateral treaties with the EU? |
| 25 | Immigration | Should sans-papiers be able to obtain a regularized residence status more easily? |
| 26 | Immigration | Are you in favor of further tightening the asylum law? |
| 27 | Immigration | Should the requirements for naturalization be increased? |
| 28 | Foreign Policy | Should Switzerland start membership negotiations with the EU? |
| 29 | Foreign Policy | Should Switzerland strive for a free trade agreement with the USA? |
| 30 | Foreign Policy | An initiative calls for liability rules for Swiss companies with regard to compliance with human rights and environmental standards abroad to be tightened. Do you support this proposal? |
| 31 | Foreign Policy | Are you in favour of Switzerland's candidacy for a seat on the UN Security Council? |

Table 9: Targets in the Politics dataset.

| ID | Domain | Target (shown below in the English form of a "topic", as given in the original dataset) |
|---|---|---|
| 1 | Society | Should same-sex couples who have registered their partnership be able to adopt children? |
| 2 | Security | Should Switzerland legalize the consumption of hard and soft drugs as well as the possession of such drugs for personal consumption? |
| 3 | Society | Would you support the right of doctors in Switzerland to help someone die with impunity? |
| 4 | Society | Would you support the introduction of a woman's quota for the Boards of Directors of listed companies? |
| 5 | Society | Switzerland has relatively strict rules when it comes to medically assisted reproduction. Should these be relaxed? |
| 6 | Security | Should young Swiss be able to choose between military service and alternative civilian service? |
| 7 | Security | The Federal Council is seeking to scale down the army from its current level of 190'000 soldiers to a level of 80'000. Do you support this idea? |
| 8 | Security | There has been an increasing tightening of rules on the acquisition and possession of weapons in recent years. Do you welcome this development? |
| 9 | Security | In the future, should juvenile criminal law place greater emphasis on longer periods of detention in closed institutions than on re-socialization measures? |
| 10 | Security | Do you think the Army should undertake policing tasks within Switzerland (e.g. protecting embassies and consulates, carrying out border protection work, and policing major events like the World Economic Forum in Davos)? |
| 11 | Security | Should the powers of the security services be increased to include "preventative" supervision of communication by post, e-mail and telephone? |
| 12 | Society | Should same-sex couples who have registered their partnership be able to adopt children? |
| 13 | Society | In June 2015 the Swiss people approved the relaxation of the rules for medically assisted reproduction (referendum on pre-implantation diagnosis, PID). Do you welcome this decision? |
| 14 | Society | Should the consumption of cannabis as well as its possession for personal use be legalised? |
| 15 | Society | Would you agree to the introduction of a minimum proportion of women as members of the board of directors or managements boards of companies listed on the stock exchange? |
| 16 | Society | Would you support the right of doctors in Switzerland to help someone die with impunity? |
| 17 | Society | Would you appreciate the introduction of automatic organ donation (presumed consent) in Switzerland? |
| 18 | Society | Do you think the federal government should withdraw from its financial support of cultural activities? |
| 19 | Security | Are you in favour of a considerable reduction of the number of soldiers to 100'000 at most? |
| 20 | Security | There has been an increasing tightening of rules on the acquisition and possession of weapons in recent years. Do you welcome this development? |
| 21 | Security | Should the powers of the security services be increased to include preventative surveillance of communication by post, e-mail and telephone? |
| 22 | Security | Should juvenile criminal law place greater emphasis on longer periods of detention in closed institutions than on re-socialization measures? |
| 23 | Security | Switzerland has one of the toughest laws against speeding. Should the law be relaxed? |
| 24 | Security | Should Switzerland terminate the Schengen Agreement with the EU and reintroduce increased identity checks directly on the border? |
| 25 | Security | Should the consumption of cannabis as well as its possession for personal use be legalised? |
| 26 | Society | Should cannabis use be legalized? |
| 27 | Security | Should Switzerland terminate the Schengen Agreement with the EU, in order to reintroduce more security checks directly on the border? |
| 28 | Society | Should same-sex couples have the same rights as heterosexual couples in all areas? |
| 29 | Society | Should the rules for reproductive medicine be further relaxed? |
| 30 | Society | Would you be in favour of a doctor being allowed to administer direct active euthanasia in Switzerland? |
| 31 | Security | Should the Federal Council's proposal to tighten the conditions for admission to the civil service be abandoned? |
| 32 | Security | Should the export of war materials from Switzerland be banned? |

Table 10: Targets in the Society dataset.