# OpenReview forum: "Cross-Lingual Cross-Target Stance Detection with Dual Knowledge Distillation Framework"
_EMNLP/2023/Conference — EMNLP 2023 Main_

### Official Review · Reviewer_UMp6 · 2023-07-29

**Soundness:** 3

**Excitement:**

3: Ambivalent: It has merits (e.g., it reports state-of-the-art results, the idea is nice), but there are key weaknesses (e.g., it describes incremental work), and it can significantly benefit from another round of revision. However, I won't object to accepting it if my co-reviewers champion it.

**Missing References:**

[1] Vamvas, Jannis, and Rico Sennrich. "X-stance: A Multilingual Multi-Target Dataset for Stance Detection." 5th SwissText & 16th KONVENS Joint Conference 2020. CEUR-WS. org, 2020.

[2] Timo Schick and Hinrich Schütze. 2021. Exploiting Cloze-Questions for Few-Shot Text Classification and Natural Language Inference. In Proceedings of the 16th Conference of the European Chapter of the Association for Computational Linguistics: Main Volume, pages 255–269, Online. Association for Computational Linguistics.

[3] Hardalov, Momchil, et al. "Few-shot cross-lingual stance detection with sentiment-based pre-training." Proceedings of the AAAI Conference on Artificial Intelligence. Vol. 36. No. 10. 2022.

[4] Valentin Barriere, Alexandra Balahur, and Brian Ravenet. 2022. Debating Europe: A Multilingual Multi-Target Stance Classification Dataset of Online Debates. In Proceedings of the LREC 2022 workshop on Natural Language Processing for Political Sciences, pages 16–21, Marseille, France. European Language Resources Association.

**Paper Topic And Main Contributions:**

This paper is about the special case of stance detection where both the language of the input and the topic (target) are zero-shot. The authors propose a knowledge distillation approach with two teachers, where one teacher is optimized for cross-lingual generalization and the other for cross-topic generalization. A student model is then trained on an interpolation of their predictions on unlabeled data in the target language.
The authors experiment with transfer from German training data to French and English test data. They find that their approach outperforms a range of baselines.

**Questions For The Authors:**

Thanks for the paper, I enjoyed reading it.

A. By what method did you select the various hyperparameters, especially K, tau and alpha? Did you use the validation set in the source language or in the target language? Are Figures 3 and 4 (which illustrate sensitivity to hyperparameter choice) based on the validation set or on the test set?

B. Imprecise claim about related work: L010f: “previous research has ignored the practical issue of no labeled data available in target language” / L059f: “current research … has not addressed the issue when there are no labeled data available in the target language” - See references [1–4] (some of which already cited), which are all concerned with zero-shot transfer to a target language.

C. What is your motivation to evaluate on two subsets of X-stance and not on the complete dataset? To me, the composition of the “Politics” and “Society” seems relatively arbitrary.

**Reasons To Accept:**

- The paper presents experiments that show a consistent improvement in accuracy over a broad range of baseline approaches.
- The experimental setup is clearly documented.

**Reasons To Reject:**

Generally, the paper presents the research in a clear manner. However, there are a few factors that limits its soundness and its potential impact on the field:
- Highly engineered approach: The proposed approach is more accurate than simply fine-tuning mBERT, but also much more complex. It might be challenging to adapt the approach to other tasks or settings. There is no indication that the code will be shared, which makes it altogether less likely that this approach will be adopted in future work.
- [Added based on author response Q1]: Hyperparameter are optimized using a validation set in the target language. Experiments on zero-shot transfer usually avoid this, because it violates the assumption that there are no annotated data in the target language.
- [Updated based on author response Q2]: The introduction exaggerates the novelty of the contribution and makes imprecise statements about the related work. In my view, it is not accurate to say that “previous research has ignored the practical issue of no labeled data available in target language” (L010f) and that “current research … has not addressed the issue when there are no labeled data available in the target language” (L059f); see references [1–4].

**Reproducibility:**

5: Could easily reproduce the results.

**Reviewer Confidence:**

4: Quite sure. I tried to check the important points carefully. It's unlikely, though conceivable, that I missed something that should affect my ratings.

**Typos Grammar Style And Presentation Improvements:**

- L312: “adjacent” -> “adjacency”

---

> ### Author Rebuttal · Authors · 2023-08-29
>
> # Response to Reviewer UMp6:
> We thank the reviewer very much for the valuable suggestions and comments. Below we respond to them in detail with further explanations and supplementary data.
>
> #### ***Q1. By what method did you select the various hyperparameters, especially K, tau and alpha? Did you use the validation set in the source language or in the target language? Are Figures 3 and 4 (which illustrate sensitivity to hyperparameter choice) based on the validation set or on the test set?***
>
> **Response**: We select hyperparameters K, tau and alpha using the following ways: (1) K represents the number of clusters in k-means. We determine the range of K according to the number of targets, and further select K on the validation set. (2) Tau is the temperature that controls the distribution smoothness of the logits in contrastive learning. In practice, tau in contrast learning is usually less than 1, so that the model pays more attention to negative samples and enhances generalization ability. We set tau as a range from 0.1 to 1, and further select them on the validation set. (3) Alpha balances cross-lingual distillation and cross-target distillation losses. We set alpha according to the degree of target inconsistency. We increase the value of alpha with target inconsistency (i.e., from All to Partial and None), making the dual distillation process pay more attention to cross-target transfer. We set alpha as a range from 0.1 to 1, and further select them on the validation set.
>
> We use the validation set in the target language to determine the values of hyperparameters. In the training process of the dual teacher models, we take the best model on the validation set in the source language as the final teacher models. In the distillation process of the student model, we take the best model on the validation set in the target language to make predictions on the test set in the target language. Figures 3 and 4 report the results on the test set in the target language, verifying the sensitivity and regularity of the model performance on different hyperparameters.
>
>
> #### ***Q2. Imprecise claim about related work: L010f: “previous research has ignored the practical issue of no labeled data available in target language” / L059f: “current research … has not addressed the issue when there are no labeled data available in the target language” - See references [1–4] (some of which already cited), which are all concerned with zero-shot transfer to a target language.***
>
> **Response**: Sorry for the confusion. Here we confine our work to cross-lingual stance detection and as we have stated in Lines 59-62 in the original paper “However, current research **on cross-lingual stance detection** has not addressed the practical issue when there are no labeled data available in the target language”. In the broader domain of cross-lingual text classification for example in Schick and Schütze, (2021) [2], it is indeed not a new issue that there are no labeled data in the target language. However, in this paper, we focus on cross-lingual stance detection and address the problems of no labeled data in target language and inconsistent targets between languages.
>
> Within cross-lingual stance detection, Hardalov et al. (2022) [3] are concerned with the zero-shot setting in the downstream experiments. However, they pre-trained the multilingual language model with additional corpora and prompt-tuned it with zero/few samples in the test set. Vamvas and Sennrich, (2020) [1] conducted zero-shot experiments with cross-lingual only setting (i.e., with the same range of targets) and cross-target only setting (i.e., within the same language) separately. Similarly, Barriere et al., (2022) [4] proposed a multilingual stance dataset for transfer learning across languages, targets and domains separately.
>
> It is worth noting that without the labeled data, target inconsistency causes the additional issue of unseen targets in target language, which in essence requires the consideration of both cross-lingual transfer and cross-target transfer. We tackle the two issues of unlabeled data and unseen targets in the target language simultaneously for cross-lingual cross-target stance detection, which is essentially a new task.
>
>
>
> #### ***Q3. What is your motivation to evaluate on two subsets of X-stance and not on the complete dataset? The composition of the “Politics” and “Society” seems relatively arbitrary.***
>
> **Response**: We construct Politics dataset and Society dataset from 4 different domains in X-stance, “Foreign Policy”, “Immigration”, “Society” and “Security”. X-stance has 10 domains altogether in which the target numbers and the target-text ratios vary in each domain. To cover more general cases, we choose 4 domains with the median numbers of targets, that is deleting the domains with too many targets or too less targets, and whose target-text ratios are similar to that of the original X-stance dataset. The detailed targets and their descriptions in Politics and Society datasets are listed in Tables 8 and 9 in Appendix.
>
>
> #### ***W1. Highly engineered approach: The proposed approach is more accurate than simply fine-tuning mBERT, but also much more complex. It might be challenging to adapt the approach to other tasks or settings.***
>
> **Response**: Due to the nature of cross-lingual cross-target stance detection task, which is inherently more complex than previously proposed cross-lingual stance detection and monolingual cross-target stance detection, we train two teacher models to learn language-related and target-oriented knowledge respectively. Regarding each individual teacher model, our method did not add extra computational complexity and the number of parameters in our method remains roughly the same as those of mBERT.
>
> Since the baseline methods in our original paper all use mBERT to encode input targets and text, we run an additional experiment to calculate the numbers of parameters in these methods and our own method. It can be seen from the table that mBERT contains about 178M parameters, JointCL has about 180M parameters with an extra prototypical graph, and the teacher-student distillation method CLKD has about two sets of 179M parameters. The last three lines report the parameter numbers of our CCSD method, which is comparative to those of the baselines including mBERT.
>
>
>
> | Method  | Teacher/Student             | Component                               | # Params   |
> |:---------|:-----------------------------|:-----------------------------------------|:------------|
> | mBERT   | -                           | mBERT                                   | ≈178M      |
> | JointCL | -                           | mBERT + prototypical graph + classifier | ≈180M      |
> | CLKD    | Teacher                     | mBERT + classifier                      | ≈179M      |
> | CLKD    | Student                     | mBERT + classifier                      | ≈179M      |
> | CCSD    | Cross-Lingual Teacher  | mBERT                                   | ≈178M      |
> | CCSD    | Cross-Target Teacher   | mBERT + target graph + classifier       | ≈180M      |
> | CCSD    | Student                | mBERT + classifier                      | ≈179M      |
>
>
> #### ***W2. There is no indication that the code will be shared, which makes it altogether less likely that this approach will be adopted in future work.***
> **Response**: We shall release the code after peer review period, and we have also added the available online access site to the revised paper.
>
>
> #### ***Missing References and Typos***:
>
> **Response**: We have added the missing references in the revised paper. We also carefully checked the whole paper, corrected the typos and improved the overall presentation. Thanks for pointing this out.

---

### Official Review · Reviewer_DhrL · 2023-08-03

**Soundness:** 4

**Excitement:**

4: Strong: This paper deepens the understanding of some phenomenon or lowers the barriers to an existing research direction.

**Missing References:**

[1] Huang, Hu, Bowen Zhang, Yangyang Li, Baoquan Zhang, Yuxi Sun, Chuyao Luo, and Cheng Peng. "Knowledge-enhanced Prompt-tuning for Stance Detection." ACM Transactions on Asian and Low-Resource Language Information Processing 22, no. 6 (2023): 1-20.

[2] Yingjie Li, Chenye Zhao, and Cornelia Caragea. "Improving stance detection with multi-dataset learning and knowledge distillation." In Proceedings of the 2021 Conference on Empirical Methods in Natural Language Processing, pp. 6332-6345. 2021.

[3] Bowen Zhang, Daijun Ding, and Liwen Jing. "How would Stance Detection Techniques Evolve after the Launch of ChatGPT?." arXiv e-prints (2022): arXiv-2212.

**Paper Topic And Main Contributions:**

This paper addresses the problem of stance detection and proposes a new task: cross-lingual cross-target stance detection. To achieve this goal, this paper designs a cross-lingual teacher and a cross-target teacher using source language data and a dual distillation process that transfers the two types of knowledge to the target language. To bridge the target inconsistency gap, cross-target teacher mines the category information via target representation learning and refinement, and generalizes it to the unseen targets via category-oriented contrastive learning. Both of these components seem to be effective in the new task.

Strengths:
- The cross-lingual cross-target stance detection task seems novel and interesting to many researchers.
- The experiments are generally well-designed and show the effectiveness of the method.
- This work tackles a useful task. The approach is reasonable and is backed with empirical results.

Weaknesses:
- Insufficient correlation between text and images makes it difficult to associate some paragraphs with figures.
- The experiments were conducted only on two stance classification dataset, out of the many available (e.g., WTWT, P-stance).

**Reasons To Accept:**

- The general idea seems novel and interesting to the community.
- The method seems to be effective.

**Reasons To Reject:**

As mentioned above,
- Some notations are not well-defined.
- Some part of the model description is not clear.
- The paper lacks sections on case study and error analysis, which are commonly found in high-quality research articles.

**Reproducibility:**

4: Could mostly reproduce the results, but there may be some variation because of sample variance or minor variations in their interpretation of the protocol or method.

**Reviewer Confidence:**

4: Quite sure. I tried to check the important points carefully. It's unlikely, though conceivable, that I missed something that should affect my ratings.

---

> ### Author Rebuttal · Authors · 2023-08-29
>
> # Response to Reviewer DhrL:
> Many thanks to the reviewer for the very helpful comments and suggestions. Below we respond to them in detail with further explanations and supplementary experimental results.
>
> #### ***W1. Insufficient correlation between text and images makes it difficult to associate some paragraphs with figures.***
>
> **Response**: We have carefully revised the figures and their corresponding description texts to enhance the correlations between paragraphs and figures. We have made the following modifications: (1) In Figure 1, replace “PLM” in cross-lingual teacher and cross-target teacher with “mBERT” and “Encoder” respectively, add more details to cross-lingual teacher, and add the arrow line indicating the connection between cross-target teacher and prompt-tuned cross-lingual teacher. (2) In Figure 2, replace “PLM” in cross-target teacher with “Encoder”, and add the illustrations of “target representation ${u}$”, “target semantic vector ${h}_i^t$”, “combined representation ${e}$” and “category representation ${r}$”.
>
>
> #### ***W2. The experiments were conducted only on two stance classification dataset, out of the many available (e.g., WTWT, P-stance).***
>
> **Response**: Thanks for the helpful comments. Due to the difficulties in acquiring the access of WT-WT and P-Stance from Twitter API, we are unable to conduct experiments on them. To further verify the effectiveness of our proposed method, we conduct experiments with two low-resource datasets as target language data, which are “czech” in Czech and “r-ita” in Italian. Dataset “czech” contains targets of “Miloš Zeman” and “Smoking Ban in Restaurants”, and “r-ita” contains the target “Constitutional Reform”, thus the target setting is None (i.e., None of the targets in source language and target language are the same). Therefore, we experiment on these four cross-lingual datasets: Politics-czech-None, Politics-rita-None , Society-czech-None, Society-rita-None, abbreviated as *P-czech-None*, *P-rita-None*, *S-czech-None* and *S-rita-None* datasets below.
>
>
> | **Method**     | **P-czech-None** | **P-czech-None** | **P-rita-None** | **P-rita-None** |
> |:---------------|:-----------------:|:-----------------:|:----------------:|:-----------------:|
> | **Measure**     | Acc               | F1                | Acc              | F1                |
> | **TAN**         | 53.01             | 48.89             | 60.69            | 50.34             |
> | **BiCond**      | 52.78             | 51.64             | 56.49            | 50.55             |
> | **CrossNet**    | 52.31             | 51.59             | 55.73            | 50.38             |
> | **JointCL**     | 53.94             | 53.39             | 65.84            | 52.31             |
> | **CLKD**        | 52.78             | 52.45             | 60.31            | 48.36             |
> | **mBERT-FT**    | 52.78             | 52.74             | 67.94            | 51.01             |
> | **mBERT-PT**    | 54.17             | 53.73             | **69.47**            | 51.57             |
> | **CCSD (Ours)** | **56.25**             | **55.89**             | 66.92            | **54.87**             |
>
>
>
> | **Method**      | **S-czech-None** | **S-czech-None** | **S-rita-None** | **S-rita-None** |
> |:---------------|:-----------------:|:-----------------:|:---------------:|:----------------:|
> | **Measure**     | Acc               | F1                | Acc             | F1               |
> | **TAN**         | 50.69             | 49.71             | 61.83           | 47.18            |
> | **BiCond**      | 50.69             | 50.69             | 63.61           | 48.62            |
> | **CrossNet**    | 51.39             | 51.09             | 63.87           | 49.29            |
> | **JointCL**     | 52.08             | 50.50             | 62.60           | 48.80            |
> | **CLKD**        | 51.39             | 49.48             | 64.12           | 50.76            |
> | **mBERT-FT**    | 51.39             | 50.36             | 61.07           | 50.08            |
> | **mBERT-PT**    | 51.39             | 51.05             | 61.07           | 51.58            |
> | **CCSD (Ours)** | **52.78**             | **52.55**             | **64.38**           | **54.54**            |
>
> It can be seen from the tables that our proposed CCSD outperforms all the baselines in these low-resource target languages and totally inconsistent target settings for cross-lingual cross-target stance detection. We shall report the above experimental results in the revised paper.
>
>
> #### ***W3. Some notations are not well-defined, and some part of the model description is not clear.***
>
> **Response**: Thanks. We have carefully proofread the whole paper and made the following modifications in Section 3: (1) Add the definitions of $\overline{t}_i$, ${u}_i$ in $\mathcal{U}=[{u}_i]$ and $C_k$ in $C=[C_k]$; (2) Revise descriptions on the cross-lingual template in Lines 241-245; (3) Supplement formulas for GAT modeling and so on. More modifications will be included in the revision.
>
>
> #### ***W4. The paper lacks sections on case study and error analysis, which are commonly found in high-quality research articles.***
>
> **Response**: We conduct case study to compare the prediction results of mBERT-FT and CCSD in Politics dataset and Partial setting. mBERT is fine-tuned with source language data and predicts the stance labels on target language data, which cannot generalize to the unseen targets (i.e., case 1, 3 and 4) well. However, our proposed method enhances its generalization ability on the unseen targets by mining target category on the target language data, enabling it to handle the task of cross-lingual cross-target stance detection.
>
> |Case  | Target | Text  | Stance |  mBERT  | CCSD  |
> |---|---|---|---|---|---|
> |1   | La Suisse devrait-elle conclure un accord de libre-échange avec les Etats-Unis?  | Trop de risques pour les standards environnementaux et sociaux...  |  Against |  Favor | Against  |
> |    |English Translation: Should Switzerland conclude a free trade agreement with the United States?| English Translation: Too many risks for environmental and social standards... |    | $\times$  |  $\checkmark$ |
> |2   |Les personnes sans-papiers devraient-elles pouvoir obtenir plus facilement un statut de séjour régularisé?   | Le système doit être équitable et être mis en œuvre au cas par cas.  | Against   | Favor  | Against  |
> | |English Translation: Should undocumented people be able to obtain regularized residence status more easily? |English Translation: The system must be fair and implemented on a case-by-case basis.   |     |  $\times$ | $\checkmark$  |
> |3   |La Confédération devrait-elle soutenir davantage les étrangères et étrangers dans leur intégration?   | L'intégration est une responsabilité individuelle, mais proposer des plateformes comme des cours de langues est un pas vers une meilleure intégration.  |Favor    |Against   | Favor  |
> |    |English Tranlsation: Should the Confederation provide more support to foreigners in their integration?    |English Translation: Integration is an individual responsibility, but offering platforms like language courses is a step towards better integration.  |   | $\times$ | $\checkmark$  |
> |4   |Êtes-vous en faveur de la candidature de la Suisse à un siège au Conseil de sécurité de l'ONU?   |Un siège au Conseil de sécurité de l'ONU est bénéfique à la Suisse, car celle-ci pourrait prendre part aux décisions contraignantes et agir en tant que médiatrice afin de maintenir la paix dans tous les états membres.   |Favor     |Against | Favor   |
> |    |English Translation: Are you in favor of Switzerland's candidacy for a seat on the UN Security Council?    |English Translation: A seat on the UN Security Council is beneficial to Switzerland, as it could take part in binding decisions and act as a mediator in order to maintain peace in all member states.   |   | $\times$ | $\checkmark$   |
>
>
> We also provide an error analysis on the three cases: (1) When text is too short, it needs additional background knowledge and reasoning process to predict stance labels. It is difficult to handle such situation only by transferring knowledge from the source language to the target language. (2) When the gap between targets in different languages (i.e., the correlation is small), such as political events and presidential candidates in different countries, our method suffers from performance degradation due to completely different distributions in data and knowledge. (3) When the language distance is large (such as German and Cezch), our method also makes more prediction mistakes. We plan to replace mBERT with a larger pre-trained language model as the base of cross-lingual teacher in future research.
>
> #### ***Missing References***:
>
> **Response**: We have added the missing references in the revised paper. Thanks for pointing this out.

---

### Official Review · Reviewer_W3Pa · 2023-08-06

**Soundness:** 3

**Excitement:**

3: Ambivalent: It has merits (e.g., it reports state-of-the-art results, the idea is nice), but there are key weaknesses (e.g., it describes incremental work), and it can significantly benefit from another round of revision. However, I won't object to accepting it if my co-reviewers champion it.

**Missing References:**

1. Schiller, Benjamin, Johannes Daxenberger, and Iryna Gurevych. "Stance detection benchmark: How robust is your stance detection?." KI-Künstliche Intelligenz (2021): 1-13.
2. Momchil Hardalov, Arnav Arora, Preslav Nakov, and Isabelle Augenstein. 2021. Cross-Domain Label-Adaptive Stance Detection. In Proceedings of the 2021 Conference on Empirical Methods in Natural Language Processing, pages 9011–9028, Online and Punta Cana, Dominican Republic. Association for Computational Linguistics.

**Paper Topic And Main Contributions:**

The paper presents an approach for cross-lingual and cross-target stance detection. The proposed framework is based on knowledge distillation from two teacher models: 1/ for language transfer, 2/ for stance target transfer, into a student pre-trained language model. The model aims to align the stance classifier's representations across the target and the source language, in order to be able to generalize to unseen targets and languages. The authors experiment with two datasets covering three languages (French, German and English).

**Questions For The Authors:**

1. How much does translating the prompt into the target language helps?

**Reasons To Accept:**

1. The proposed method models both the target and the language, which is indeed an important step for cross-lingual stance detection.
2. The student model can be in theory smaller in size compared to the teachers which makes it more efficient, even though the authors did not test this setup.
3. The proposed approach outperforms strong BERT-based baselines ins all settings. The authors ablate all of the model's components showing that they are important for the end prediction.

**Reasons To Reject:**

1. The proposed method is computationally expensive -- it is based on two teacher networks (with additional architecture modification like GAT) that have to be pre-trained on the source language, and then distilled back to a target langauge network.
2. The study covers only three languages, non of which is truly low-resource when in terms of pre-training data included as part of the language model's extensive training. Moreover, two of the languages come from the same dataset and domain. Hardalov et al. (2022) can be a good source of candidate datasets, as they used 16 datasets for their experiments.
3. The evaluation setup and experiments are limited in two way: 1/ to only two stance labels (in favor and against), whereas most of the existing datasets have many more, and they often do not align well in their label definitions, which can pose additional challenges for the proposed approach; 2/ pre-training on English datasets had shown good zero-shot/few-shot transfer performance, this is another baseline that should be considered by the authors.

**Reproducibility:**

4: Could mostly reproduce the results, but there may be some variation because of sample variance or minor variations in their interpretation of the protocol or method.

**Reviewer Confidence:**

4: Quite sure. I tried to check the important points carefully. It's unlikely, though conceivable, that I missed something that should affect my ratings.

---

> ### Author Rebuttal · Authors · 2023-08-29
>
> # Response to Reviewer W3Pa:
> Many thanks to the reviewer for the valuable comments and suggestions. Below we respond to them in detail with further explanations and supplementary experimental results.
>
> #### ***Q1. How much does translating the prompt into the target language help?***
>
> **Response**: We aim to strengthen the cross-lingual ability of the cross-lingual teacher model using the source language data only. To achieve this, we translate the prompt words into the target language to obtain the cross-lingual template, and prompt-tune the cross-lingual teacher with consistency constraint. Specifically, we devise consistency loss to constrain the prediction distributions between the original template and cross-lingual template, so that the model can learn language-related knowledge and enhance its cross-lingual ability.
>
> Take the Politics dataset as an example. Below we provide the quantitative analysis results under three different target settings on the two variants of the cross-lingual teacher: (a) removing consistency constraint and (b) without translating the prompt words.
>
> |Variant                       | Politics | All  |      |     | Politics | Partial |       |       | Politics  | None |       |        |
> |:-------------------------------|----------|-------|-------|-------|----------|----------|-------|-------|-----------|-------|-------|--------|
> | **Measure**| Acc      | F1    | ∆Acc  | ∆F1   | Acc      | F1       | ∆Acc  | ∆F1   | Acc       | F1    | ∆Acc  | ∆F1    |
> | **CCSD**                    | **70.11**    | **69.92** | -     | -     | **67.44**    | **67.32**    | -     | -     | **65.72**     | **65.47** | -     | -      |
> | **(a) w/o consistency**  | 68.75    | 68.37 | -1.36 | -1.55 | 66.09    | 66.06    | -1.35 | -1.26 | 62.26     | 62.14 | -3.46 | -3.33  |
> | **(b) w/o translating prompt** | 67.59    | 67.25 | -2.52 | -2.67 | 65.12    | 65.07    | -2.32 | -2.25 | 62.26     | 61.34 | -3.46 | -4.13  |
>
> The results show that both removing the consistency constraint and removing prompt word translation can lead to performance drop under all the target settings. And removing translating prompts causes more severe performance decrease (-2.67%, -2.25% and -4.13% in F1 score under three different settings). We shall report the above ablation results in the revised paper.
>
>
> #### ***W1. The proposed method is computationally expensive with two teacher networks (with additional architecture modification like GAT) pre-trained on the source language and then distilled back to a target language network.***
>
> **Response**: Due to the nature of cross-lingual cross-target stance detection task, which is inherently more complex than previously proposed cross-lingual stance detection and monolingual cross-target stance detection, we train two teacher models to learn language-related and target-oriented knowledge respectively. Regarding each individual teacher model, our method did not add extra computational complexity and the number of parameters in our method remains roughly the same as those of mBERT.
>
> Since the baseline methods in our original paper all use mBERT to encode input targets and text, we run an additional experiment to calculate the numbers of parameters in these methods and our own method. It can be seen from the table that mBERT contains about 178M parameters, JointCL has about 180M parameters with an extra prototypical graph, and the teacher-student distillation method CLKD has about two sets of 179M parameters. The last three lines report the parameter numbers of our CCSD method, which is comparative to those of the baselines including mBERT.
>
>
>
> | Method  | Teacher/Student             | Component                               | # Params   |
> |:---------|:-----------------------------|:-----------------------------------------|:------------|
> | mBERT   | -                           | mBERT                                   | ≈178M      |
> | JointCL | -                           | mBERT + prototypical graph + classifier | ≈180M      |
> | CLKD    | Teacher                     | mBERT + classifier                      | ≈179M      |
> | CLKD    | Student                     | mBERT + classifier                      | ≈179M      |
> | CCSD    | Cross-Lingual Teacher  | mBERT                                   | ≈178M      |
> | CCSD    | Cross-Target Teacher   | mBERT + target graph + classifier       | ≈180M      |
> | CCSD    | Student                | mBERT + classifier                      | ≈179M      |
>
>
>
> #### ***W2. The study covers only three languages, non of which is truly low-resource when in terms of pre-training data included as part of the language model's extensive training. Moreover, two of the languages come from the same dataset and domain. Hardalov et al. (2022) can be a good source of candidate datasets, as they used 16 datasets for their experiments.***
>
> **Response**: Many thanks for the valuable suggestions. We construct *Politics* dataset and *Society* dataset from 4 different domains in X-stance, “Foreign Policy”, “Immigration”, “Society” and “Security”. X-stance has 10 domains altogether in which the target numbers and the target-text ratios vary in each domain. To cover more general cases, we choose 4 domains with the median numbers of targets and whose target-text ratios are similar to that of the original X-stance dataset. The detailed targets and their descriptions in Politics and Society datasets are listed in Tables 8 and 9 in Appendix.
>
> We conduct additional experiments on two low-resource language datasets “czech” in Czech and “r-ita” in Italian from Hardalov et al. (2022). Dataset “czech” contains targets of “Miloš Zeman” and “Smoking Ban in Restaurants”, and “r-ita” contains the target “Constitutional Reform”, thus the target setting is None (i.e., None of the targets in source language and target language are the same). Therefore, we experiment on these four cross-lingual datasets: Politics-czech-None, Politics-rita-None , Society-czech-None, Society-rita-None, abbreviated as *P-czech-None*, *P-rita-None*, *S-czech-None* and *S-rita-None* datasets below.
>
>
> | **Method**     | **P-czech-None** | **P-czech-None** | **P-rita-None** | **P-rita-None** |
> |:---------------|:-----------------:|:-----------------:|:----------------:|:-----------------:|
> | **Measure**     | Acc               | F1                | Acc              | F1                |
> | **TAN**         | 53.01             | 48.89             | 60.69            | 50.34             |
> | **BiCond**      | 52.78             | 51.64             | 56.49            | 50.55             |
> | **CrossNet**    | 52.31             | 51.59             | 55.73            | 50.38             |
> | **JointCL**     | 53.94             | 53.39             | 65.84            | 52.31             |
> | **CLKD**        | 52.78             | 52.45             | 60.31            | 48.36             |
> | **mBERT-FT**    | 52.78             | 52.74             | 67.94            | 51.01             |
> | **mBERT-PT**    | 54.17             | 53.73             | **69.47**            | 51.57             |
> | **CCSD (Ours)** | **56.25**             | **55.89**             | 66.92            | **54.87**             |
>
>
>
> | **Method**      | **S-czech-None** | **S-czech-None** | **S-rita-None** | **S-rita-None** |
> |:---------------|:-----------------:|:-----------------:|:---------------:|:----------------:|
> | **Measure**     | Acc               | F1                | Acc             | F1               |
> | **TAN**         | 50.69             | 49.71             | 61.83           | 47.18            |
> | **BiCond**      | 50.69             | 50.69             | 63.61           | 48.62            |
> | **CrossNet**    | 51.39             | 51.09             | 63.87           | 49.29            |
> | **JointCL**     | 52.08             | 50.50             | 62.60           | 48.80            |
> | **CLKD**        | 51.39             | 49.48             | 64.12           | 50.76            |
> | **mBERT-FT**    | 51.39             | 50.36             | 61.07           | 50.08            |
> | **mBERT-PT**    | 51.39             | 51.05             | 61.07           | 51.58            |
> | **CCSD (Ours)** | **52.78**             | **52.55**             | **64.38**           | **54.54**            |
>
>
> It can be seen from the tables that our proposed CCSD outperforms all the baselines in these low-resource target languages and totally inconsistent target settings for cross-lingual cross-target stance detection. We shall report the above experimental results in the revised paper.
>
> #### ***W3. The evaluation setup and experiments are limited in two way: (1) to only two stance labels (in favor and against), whereas most of the existing datasets have many more, and they often do not align well in their label definitions, which can pose additional challenges for the proposed approach; (2) pre-training on English datasets had shown good zero-shot/few-shot transfer performance, this is another baseline that should be considered by the authors.***
>
> **Response**: This is a very good point. Although misalignment of labels between different datasets is common in stance detection, this important issue has rarely been considered in cross-lingual stance detection. The pioneer work of Hardalov et al.,(2021) in EMNLP 2021 tackled this issue in monolingual setting by devising label embedding in domain adversarial training. In cross-lingual stance detection domain, Hardalov et al., (2022) in AAAI 2022 proposed label embedding with pre-training to alleviate the situation when labels do not align well in target language. As our problem definition of cross-lingual cross-target stance detection assumes that the labeled data are available only in source language, we consider to tackle this additional research challenge by cross-lingual data augmentation in our future work.
>
>
> ChatGPT as a generative large language model mainly pre-trained with English as well as other languages has shown good zero-shot transfer performance, we conduct additional experiments and compare our proposed method with ChatGPT in zero-shot setting (with GPT-3.5-turbo-0613). The prompt used in the experiment is “La position de <TEXT> envers <TARGET> est? Select one word from ‘favor’ and ‘against’.”. It can be seen from the tables that our proposed CCSD outperforms ChatGPT in both datasets under all the settings for cross-lingual cross-target stance detection. Compared with the results of mBERT-PT, ChatGPT performs better in "Politics-Partial", "Politics-None" and "Society-None". We shall report the above experimental results in the revised paper.
>
>
>
> | **Method**              | **Politics-All** | **Politics-All** | **Politics-Partial** | **Politics-Partial** | **Politics-None** | **Politics-None** |
> |:-----------------------|:----------------:|:----------------:|:--------------------:|:--------------------:|:-----------------:|:------------------:|
> | **Measure**             | Acc              | F1               | Acc                  | F1                   | Acc               | F1                 |
> | **ChatGPT (zero-shot)** | 65.95            | 65.41            | **68.02**                | **67.63**                | 61.32             | 60.30              |
> | **mBERT-PT**            | 67.82            | 67.76            | 66.28                | 65.90                | 57.86             | 57.48              |
> | **CCSD (Ours)**         | **70.11**            | **69.92**            | 67.44                | 67.32                | **65.72**             | **65.47**              |
>
>
> | **Method**              | **Society-All** | **Society-All** | **Society-Partial** | **Society-Partial** | **Society-None** | **Society-None** |
> |:-----------------------|:---------------:|:---------------:|:-------------------:|:-------------------:|:----------------:|:-----------------:|
> | **Measure**             | Acc             | F1              | Acc                 | F1                  | Acc              | F1                |
> | **ChatGPT (zero-shot)** | 59.46           | 58.63           | 56.90               | 56.38               | **62.96**            | 59.97             |
> | **mBERT-PT**            | 66.82           | 66.31           | 64.37               | 63.84               | 61.73            | 59.00             |
> | **CCSD (Ours)**         | **67.87**           | **67.32**           | **66.09**               | **65.43**               | **62.96**            | **62.48**             |
>
>
>
>
>
> #### ***Missing References***:
>
> **Response**: We have added the missing references in the revised paper. Thanks for pointing this out.

---

### Meta-Review · Area_Chair_zriZ · 2023-09-19

**Recommendation:** 4

**Metareview:**

Reviewers are cautiously positive in their assesment of the paper (3-4).  There are objections to the task being suboptimally selected relative to other work in cross-lingual stance detection and to the set-up being unnecessarily complex.  Despite this, reviewers agree that the experimental set-up is well-documented and the results are positive.

---

### Decision · Program_Chairs · 2023-10-07

**Decision:**

Accept-Main

**Comment:**

Reviewers are cautiously positive in their assesment of the paper (3-4).  There are objections to the task being suboptimally selected relative to other work in cross-lingual stance detection and to the set-up being unnecessarily complex.  Despite this, reviewers agree that the experimental set-up is well-documented and the results are positive.